# Comparison of Parametric and Nonparametric Methods for Estimating Size–Density Relationships in Old-Growth Japanese Cedar (*Cryptomeria japonica* D. Don) Plantations

**Chyi-Rong Chiou [1], Ching-Peng Cheng [1],\* and Sheng-I Yang [2]** 

[1]  School of Forestry and Resource Conservation, National Taiwan University, No 1, Section 4, Roosevelt Rd., Taipei 106, Taiwan; esclove@ntu.edu.tw

[2]  Department of Forestry, Wildlife and Fisheries, University of Tennessee, 274 Ellington Plant Sciences Building, Knoxville, TN 37996-4563, USA; syang47@utk.edu

\*  Correspondence: d03625003@ntu.edu.tw; Tel.: +886-223-697-658

**Abstract:** Accurately quantifying the size–density relationships is important to predict stand development, estimate stand carrying capacity and prescribe silvicultural treatments. Parametric methods, such as segmented regression, were proposed to estimate the complicated trajectory of size–density relationships. However, applying nonparametric methods to assess stand development has not been explicitly examined. In this study, we compared parametric and nonparametric methods for estimating size–density relationships for Japanese cedar plantations in Taiwan. Specifically, we compared the efficacy of two segmented regression models with the penalized spline and random forest for regression methods. We also examined various stages in stand development for old-growth Japanese cedar stands. Data collected from 237 Japanese cedar permanent plots were used in model fitting and validation. Results indicated that the parametric and nonparametric methods used in this study can provide reliable estimates of the size–density relationship for Japanese cedar. Higher accuracy was achieved before the stands diverged from the self-thinning line. The penalized spline approach behaved consistently well regardless of datasets or stages in stand development, while the predictability of the random forest algorithm slightly decreased when the validation data was fitted. The results of this study provide insights on the use of methods to quantify the size–density relationships as well as enhance the understanding of long-term stand development.

**Keywords:** segmented regression; penalized spline; random forest; number of trees per unit area; quadratic mean diameter

## 1. Introduction

Accurately quantifying size–density relationships is important in forest management to predict stand development, estimate stand carrying capacity and prescribe silvicultural treatments [1,2]. The relationship between number of trees per unit area and the quadratic mean diameter ($D_q$) of a stand on a log-log scale is commonly used to describe the size–density relationship of a stand. This relationship generally consists of two major stages: the density-independent mortality stage (Stage I) and the competition-induced mortality stage (Stage II) [3]. Stage I is represented by pre-canopy conditions where mortality of trees is independent of stand density. With the limited site resources and growing space, stand transfers from Stage I to Stage II due to occurrence of competition between trees, lead to the decrease in number of trees per unit area (i.e., competition-induced mortality). Most forest stands experience competition but the extent of competition varies among regions, genotypes,

species, silvicultural treatments, environmental and climatic conditions [4–7]. Later, the number of trees per unit area and the average tree size in the stand will reach and then stay on a relatively stable equilibrium for a period of time. This period is also known as maximum size–density relationship, or self-thinning [1]. Eventually, the number of trees per unit area will decrease with minor or no increase of average tree size since residual neighboring trees are not able to fully compensate the canopy gaps [8].

In the past, many studies focused on examining the maximum size–density relationships (i.e., self-thinning) [5,9–11]. Reineke [10] found that for various species, the limiting relationship between logarithm of number of trees per unit area and logarithm of quadratic mean diameter is linear, which has been widely used in forest density management. Similarly, Yoda et al. [12] indicated the mean mass (or weight) and number of plants per unit area on the log-log scale follow the -3/2 rule of self-thinning when the stands are in the competition-induced mortality stage. Hagihara [9] derived the relationship between the competition-density index proposed by Kira et al. [13] and Yoda's self-thinning rule. Rather than focusing on the self-thinning behavior, segmented regression as a parametric method was proposed to estimate the complete trajectory of size–density relationship [3,4,14]. VanderSchaaf and Burkhart [3] examined the effect of initial planting density on size–density relationships using four- and two-segment regression models for loblolly pine. Cao and Dean [4] found that the curvilinear trajectories of the size–density relationships for slash pine can be reliably predicted using segmented regression models. In the past decades, due to an increase in computing power, nonparametric methods with a computation intensive algorithm have been widely applied and proved useful to address problems in forestry [15–18]. However, to our knowledge, applying nonparametric methods for estimating size–density relationship trajectory has not been explicitly examined, especially for old growth forests.

Japanese cedar (*Cryptomeria japonica* D. Don), native in Japan, is one of the major timber species due to its desirable wood properties. Nishizono and Tanaka [19] evaluated the stand development in unthinned and thinned old-growth Japanese cedar plantations in northeastern Japan. It was indicated that the thinning treatment has an obvious effect on the long-term stand development. Ogawa [20] found that the self-thinning line of 21-year old Japanese cedar in central Japan followed Yoda's -3/2 self-thinning rule. Ogawa and Hagihara [21] indicated that the mortality of trees occurred randomly after the self-thinning phase. In the 19[th] century, Japanese cedar was introduced to Taiwan, and is one of the major plantation species in Taiwan. Similar to Japan, Japanese cedar was widely planted throughout the mountainous region of the Taiwan Island. However, due to subtropical climate and Pacific Ocean location, the temperature and humidity in Taiwan are much higher than those in northern Japan. Yang [22] reported that the growth of Japanese cedar in Taiwan was faster than that in the northern Japan. In this study, we compared parametric and nonparametric methods for estimating the size–density relationship for Japanese cedar plantations in Taiwan. Specifically, we compared the efficacy of two segmented regression models proposed by VanderSchaaf and Burkhart [3] with the penalized spline and random forest for regression methods. Data collected by Taiwan Forestry Bureau and National Taiwan University Experimental Forest were used in analyses. The age of some Japanese cedar plantations in National Taiwan University Experimental Forest has exceeded a hundred years. The results of this study provide new insights regarding the methods used to quantify size–density relationships and enhance the understanding of long-term stand development. With rising concerns about climate change and global warming, this study conducted in the subtropical island (Taiwan) will be beneficial for long-term Japanese cedar plantation management in the temperate zone, like Japan.

## 2. Materials and Methods

### 2.1. Data

#### 2.1.1. National Forest Resource Inventory Permanent Plots

Data were obtained from the National Forest Resource Inventory (NFRI), which is administered by the Taiwan Forestry Bureau. In 1997, Japanese cedar permanent plots with a size of 0.02 or 0.05 ha were installed in the eight Forest District Offices (Luodong, Hsinchu, Dongshi, Nantou, Chiayi, Pingtung, Taitung and Hualien) across the entire island (see Figure 1). The plantations selected for plot installation were only planted with Japanese cedar. Each Office followed the identical standard of the silvicultural treatments and forest management regardless of sites or elevations. Herbaceous vegetations were removed in the first six years, and seedlings were replanted in the third year to ensure 80% survivals. No other treatments were applied on the plots. The number of random permanent plots that were established in each Forest District Office is proportional to the total area of Japanese cedar plantations in each office [23]. Trees with diameter at breast height (DBH) greater than 6 cm were remeasured every five years during 1997–2009. Measurements, including DBH, total tree height, crown condition indicator (dimensions) and site condition, were recorded. Plots with a majority of poor-quality standing trees or dominated by other weedy species were not included in analyses. A total number of 515 observations were collected from 222 permanent plots where the Japanese cedar basal area was greater than 30% of the total basal area and Japanese cedar dominated at least 20% of the main canopy. The total number of sample trees ranged from 10 to 88 trees per plot at stand ages of 8–93 (years).

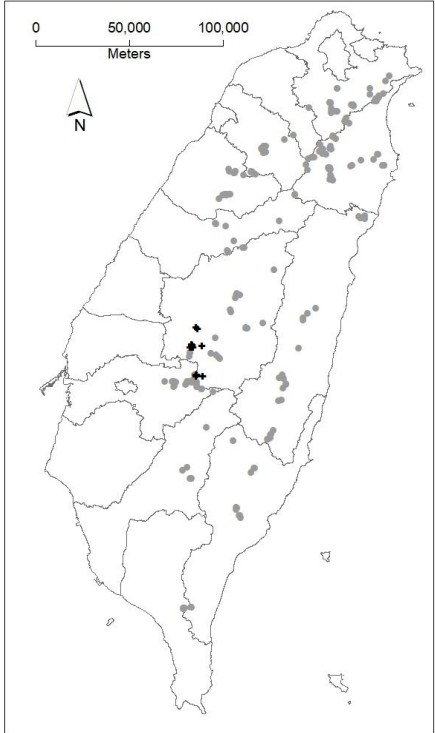

**Figure 1.** Locations of Japanese cedar long-term growth experiments (black plus signs) and nationwide permanent plots (gray dots) in Taiwan.

#### 2.1.2. Long-Term Experimental Plots

Fifteen long-term experimental plots for the growth study of Japanese cedar maintained by the National Taiwan University Experimental Forest (NTUEF) were used. NTUEF is located at the west side of the Central Mountain Range of the island. The plots were established between 1921

and 1945, with plot size ranging from 0.08 to 0.17 ha at the elevation of 750–2300 m above sea level. The distribution of the plots is shown in Figure 1. The average annual temperature of 15.5–24.6 °C and average precipitation of 1600–2700 mm were recorded during 1941–2005 [24]. Measurements of live trees, including DBH and total tree height, were taken every 1–5 years.

Quadratic mean diameter ($D_q$) and number of trees per ha (TPH) were computed. Stand characteristics are summarized in Table 1.

**Table 1.** A summary of stand characteristics for the old-growth Japanese cedar plantations in Taiwan. # of obs. is total number of observations, Age is stand age in years, DBH is diameter at breast height in cm, $D_q$ is quadratic mean diameter, TPH is trees per hectare and BA is basal area ($m^2$) per hectare.

| Sources | # of Plots | # of Obs. | Stand Variable | Average | Min. | Max. | Std. |
|---|---|---|---|---|---|---|---|
| Long-term experimental plots | 15 | 460 | Age | 36 | 2 | 105 | 24 |
| | | | DBH | 26.6 | 1.1 | 56.1 | 10.4 |
| | | | $D_q$ | 27.4 | 1.1 | 58.8 | 10.6 |
| | | | TPH | 1269 | 190 | 3657 | 693 |
| | | | BA | 60.5 | 0.2 | 170.2 | 26.2 |
| NFRI permanent plots | 222 | 515 | Age | 43 | 8 | 93 | 14 |
| | | | DBH | 25.9 | 11.9 | 55.6 | 8.2 |
| | | | $D_q$ | 27.2 | 12.1 | 57.4 | 7.6 |
| | | | TPH | 1316 | 340 | 4400 | 604 |
| | | | BA | 79.5 | 13.3 | 197.6 | 35 |

## 2.2. Parametric and Nonparametric Methods

### 2.2.1. Segmented Regression Models

Two segmented regression models used in this study were proposed by VanderSchaaf and Burkhart [3]. VanderSchaaf and Burkhart [3] noted that stand development in the Stage II competition-induced mortality stage can be shown in three phases: the stand approaching the self-thinning line phase (Phase I), within the self-thinning line phase (Phase II) and divergence from self-thinning line phase (Phase III). Thus, the full model including two stages and three phases was written as:

$$LnN = \beta_1 I_1 + \left[\beta_1 + \beta_2\left(LnD_q - \alpha_1\right)^2\right]I_2 + \left[\beta_1 + \beta_2(\alpha_2 - \alpha_1)^2 + \beta_3\left(LnD_q - \alpha_2\right)\right]I_3$$
$$+ \left[\beta_1 + \beta_2(\alpha_2 - \alpha_1)^2 + \beta_3(\alpha_3 - \alpha_2) + \beta_4\left(LnD_q - \alpha_3\right)\right]I_4 \tag{1}$$

where Ln = natural logarithm;
  N = number of trees per ha;
  Dq = quadratic mean diameter (cm);
  $I_1$ = 1 if $LnDq \leq \alpha_1$, 0 otherwise;
  $I_2$ = 1 if $LnDq > \alpha_1$ and $LnDq \leq \alpha_2$, 0 otherwise;
  $I_3$ = 1 if $LnDq > \alpha_2$ and $LnDq \leq \alpha_3$, 0 otherwise;
  $I_4$ = 1 if $LnDq > \alpha_3$, 0 otherwise;
  $\beta_1, \beta_2, \beta_3, \beta_4$ = coefficient;
  $\alpha_1$ = joint point between Stage I and Stage II;
  $\alpha_2$ = joint point between Phase I and Phase II;
  $\alpha_3$ = joint point between Phase II and Phase III.
  The second model is a reduced form of the first model, which is

$$LnN = \beta_1 I_1 + \left[\beta_1 + \beta_2\left(LnD_q - \alpha_1\right)^2\right]I_2 \tag{2}$$

where all symbols are as previously defined. The full model includes seven parameters while the reduced model has only three parameters. Compared to the full model, the size–density relationships

in the Stage II (i.e., competition-induced mortality stage) were described by a single quadratic function in the reduced model. In the full model, a linear self-thinning phase was assumed while in the reduced model, the self-thinning phase was curvilinear. In addition, the divergence from the self-thinning phase (i.e., Phase III) was depicted as a line in the full model, whereas it was depicted as a curve in the reduced model.

### 2.2.2. Penalized Spline

Penalized spline as a type of spline regression includes two main components: a smooth function and a roughness penalty term (i.e., regularization term) [25]. The smooth function is applied to control the smoothness of the joint functions, which is determined by number of knots in the function. Similar to ridge regression, the roughness penalty term is used to shrink the coefficients, which is controlled by the smoothing parameter ($\lambda$). In this study, a cubic spline function was used:

$$LnN = f(\text{LnDq}) = \beta_1 + \beta_2 \left(\text{LnD}_q\right)^2 + \beta_3 \left(\text{LnD}_q\right)^3 + \sum_{i=1}^{T} b_i \left(LnD_q - k_i\right)^3_+ \tag{3}$$

where $b_i$ = the weight for $i^{th}$ function;
   $k_i$ = the position of knot i;
   $\left(LnD_q - k_i\right)^3_+ = \left(LnD_q - k_i\right)^3$ if $LnD_q > k_i$, 0;
   other symbols are as previously defined.
   The coefficients were estimated by minimizing

$$L(\text{LnDq}) = \sum_{j=1}^{n} \left[LnN_j - f\left(LnD_{q_j}\right)\right]^2 + \lambda \int f''\left(\text{LnD}_q\right) d\left(LnD_q\right) \tag{4}$$

where $\lambda \int f''\left(\text{LnD}_q\right) d\left(LnD_q\right)$ is the roughness penalty term. In this study, number of knots and the smoothing parameter were selected by the minimum generalized cross validation (GCV) value [26].

### 2.2.3. Random Forest for Regression

Random forest for regression is an ensemble of decision trees algorithm at the training stage proposed by Breiman [27]. The ensemble algorithm can greatly reduce the instability of prediction from a single decision tree [28]. In random forest algorithm, the selection of both observations and variables are random. In this study, $LnD_q$ and $(LnD_q)^2$ are included as independent variables. Each variable has an equal chance of being sampled at each split. Hyperparameters, including number of decision trees and node size, were selected by using the minimum mean square error using 10-fold cross validation simulated 1000 times.

### 2.3. Parameter Estimation and Method Evaluation

In this study, long-term experimental plots installed in the National Taiwan University Experimental Forest (NTUEF) were used as a fitting dataset. The National Forest Resource Inventory (NFRI) permanent plots were utilized for model validation. Commonly, the validation dataset was a random subset of the data. However, in this study, it is more appropriate to keep the data separated in model fitting and validation because the data were obtained from two different sources. The two fully independent datasets were advantageous to evaluate the robustness of the models. Although the number of the NTUEF long-term experimental plots is less than that of the NFRI permanent plots, more remeasurements per plot were taken. Thus, the NTUEF plots were more suitable in model fitting to capture the complete size–density relationship trajectories.

The coefficients in the segmented regression models were estimated using the package minpack.lm, while penalized spline and random forest were implemented using the packages splines and randomForest, respectively, in R.

To compare the overall performance and the estimation in each stage among various models, mean bias (MB) and root mean square error (RMSE) were calculated as:

$$\text{MB} = \frac{1}{n} \sum_{i=1}^{n} (\hat{y}_i - y_i) \tag{5}$$

$$\text{RMSE} = \left[ \frac{1}{n} \sum_{i=1}^{n} (\hat{y}_i - y_i)^2 \right]^{\frac{1}{2}} \tag{6}$$

where n = number of plots in the dataset;

$y_i$ = logarithm of observed number of trees per ha (*LnN*);

$\hat{y}_i$ = logarithm of predicted number of trees per ha (*LnN̂*).

## 3. Results and Discussion

### 3.1. Size–Density Relationships Estimated by the Segmented Regression Models

As shown in Figure 2, the trend of the size–density relationships predicted by the full and reduced models is similar to that of the observed stand development trajectory. Both models can be used to accurately describe the stand development with low bias for the training and validation data (see Table 3). Based on the estimated coefficients of $\alpha_1$ in both models, the stands transferred from Stage I to Stage II (i.e., initiation of competition-induced mortality) when quadratic mean diameter approached between 6.25 and 10.9 cm (LnD$_q$ between 1.833 and 2.389). In Table 2, it shows that the coefficients of $\alpha_2$ and $\beta_2$ are not statistically significant from zero, which indicates that the transition between Phase I and Phase II in the competition-induced mortality stage is not obvious in the study. Notably, the reduced model well describes the curvilinear trajectory of LnN-LnDq relationship over the entire range of the Stage II.

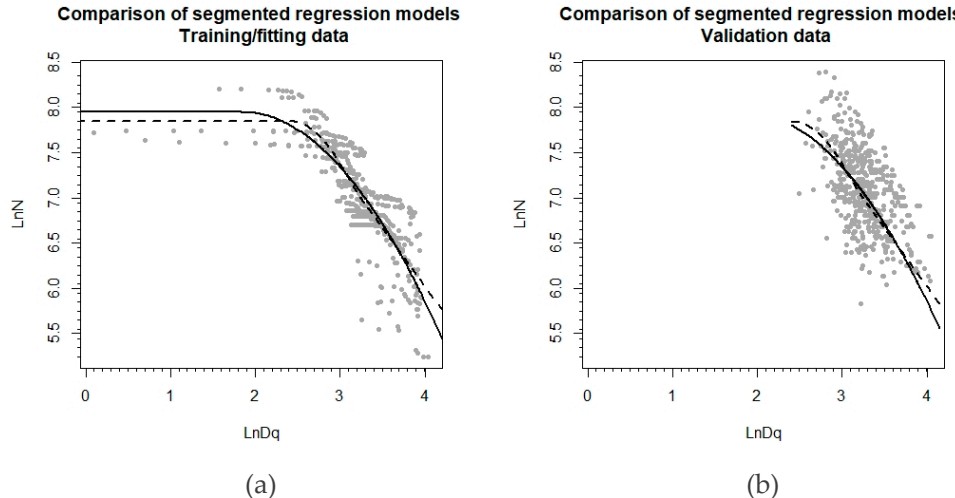

**Figure 2.** Prediction of size–density relationships by the full (dashed line) and reduced (solid line) segmented regression models proposed by VanderSchaaf and Burkhart [3]. The gray dots represent the observations. (**a**) Data collected from the long-term experimental plots installed in the National Taiwan University Experimental Forest (NTUEF) were used as a fitting dataset, while (**b**) those collected from the National Forest Resource Inventory (NFRI) permanent plots were utilized for model validation.

**Table 2.** A summary of coefficient estimation of the full and reduced segmented regression models proposed by VanderSchaaf and Burkhart [3].

| Coefficient | Full Model | | Reduced Model | |
|:---:|:---:|:---:|:---:|:---:|
| | **Estimate** | **$p$-Value** | **Estimate** | **$p$-Value** |
| $\beta_1$ | 7.850 | <0.0001 * | 7.949 | <0.0001 * |
| $\beta_2$ | −1.267 | 0.079 | −0.445 | <0.0001 * |
| $\beta_3$ | −1.579 | 0.008 * | | |
| $\beta_4$ | −1.293 | <0.0001 * | | |
| $\alpha_1$ | 2.389 | <0.0001 * | 1.833 | <0.0001 * |
| $\alpha_2$ | 3.027 | 0.225 | | |
| $\alpha_3$ | 3.211 | <0.0001 * | | |

* $p$-value less than 0.05 was highlighted.

The slope of the self-thinning line estimated was −1.579, which was obtained from the coefficient $\beta_3$ of the full model (see Table 2). Nishizono and Tanaka [19] examined the relationship between quadratic mean diameter and number of trees per ha of Japanese cedar plantations in northeastern Japan. The slope of −1.486 was estimated using data from unthinned long-term experimental plots. The slope found in this study is steeper than that reported by Nishizono and Tanaka [19], which may be due to the difference in environmental conditions and management practices between two regions. It was reported that the growth rate of Japanese cedar in Taiwan is higher than that in northern Japan [22]. Due to the whole-year warm and humid weather in Taiwan, the longer growing season may cause a higher competition intensity among individuals. The variation of self-thinning lines among regions was also found for other conifer species. Zhang et al. [29] reported that the slope of self-thinning line for ponderosa pine (*Pinus ponderosa*) in California is consistently greater than that in Rocky Mountains. Zhang et al. [7] indicated that temperature is a primary factor to affect the self-thinning trajectories for Chinese fir plantations. Among four provinces in China, the areas with greater annual precipitation and summer mean maximum temperature had a significantly steeper slope of the self-thinning line for Chinese fir.

Reineke [10] proposed the stand density index (SDI) to evaluate carrying capacity of a stand, which is defined as:

$$SDI = N\left[\frac{D_q}{25.4}\right]^b \tag{7}$$

where b is the slope of the self-thinning line. As indicated by VanderSchaaf and Burkhart [3], the maximum SDI can be estimated using the LnN and $LnD_q$ at the beginning and ending points of the self-thinning line. The resultant maximum SDI of 1190–2126 was calculated in this study. Fukumoto et al. [30] found that SDI is an important predictor in the individual-level distance-independent diameter growth models, which improves the predictability of tree diameter growth of Japanese cedar under different thinning intensities.

In short, both segmented regression models can be used to predict the long-term stand development for Japanese cedar. However, it should be noted that both models deployed in this study include a linear segment to describe the size–density relationships in Stage I where mortality is not evident. If self-thinning has obviously occurred in the stands during the first measured period, a horizontal line of the first segment in the model may not be sufficient to predict the decrease of stand density. Thus, Cao and Dean [4] suggested a quadratic-quadratic segmented model to handle this situation, which nullifies the initial horizontal segment. Although both models can produce reliable estimates of size–density relationships, the reduced model is recommended because the distinct three phases in the competition-induced mortality stage (i.e., Stage II) may not be obvious for every dataset. VanderSchaaf and Burkhart [3] found that convergence was not obtained for two of the nine planting densities when the full model was fitted due to a lack of sufficient self-thinning. With less parameters, the reduced model can still capture the complicated trajectory well. A parsimonious model is preferable

when identifying the initial values in model fitting. The second segment of the reduced model can be modified if the size–density relationship is linear in the Stage II.

### 3.2. Comparison of Parametric and Nonparametric Methods

In general, the size–density relationships for Japanese cedar could be reliably predicted by the parametric and nonparametric methods used in this study. Results indicate that the estimates of number of trees per ha are close to the observed stand density. As shown in Table 3, the MB were between −0.001 and 0.110, and RMSE were between 0.040 and 0.190. The predictability of the methods was evaluated at different stages or phases of the stand development. Table 4 showed that higher accuracy was achieved when estimating the $LnN - LnD_q$ relationships before the stands diverged from the self-thinning line (i.e., Phase III in Stage II). Overall, the performance of the segmented regression models is slightly worse than that of penalized spline or random forest for regression methods when the fitting data were fitted. However, for the validation dataset, the size–density relationships were better predicted by the segmented regression models (see Table 3), which implied that the segmented models could be relatively more robust than the penalized spline and random forest algorithm. Notably, the random forest for regression method yields the lowest RMSE for the training dataset, while it has the highest RMSE when estimating stand density in the validation data (see Table 3). Compared to other three methods used, the penalized spline approach behaved consistently well regardless of datasets or stand development stages.

**Table 3.** A summary of overall mean bias (MB) and root mean square error (RMSE) for parametric and nonparametric methods. Data collected from the long-term experimental plots installed in the National Taiwan University Experimental Forest (NTUEF) were used as a fitting dataset, while those collected from the National Forest Resource Inventory (NFRI) permanent plots were utilized for model validation.

| Method | Fitting Data | | Validation Data | |
|---|---|---|---|---|
| | MB | RMSE | MB | RMSE |
| Segmented Regression (Full) | 0.000 | 0.079 | 0.098 | 0.153 |
| Segmented Regression (Reduced) | 0.000 | 0.083 | 0.083 | 0.151 |
| Penalized Spline | 0.000 | 0.075 | 0.103 | 0.158 |
| Random Forest for Regression | −0.001 | 0.040 | 0.110 | 0.190 |

**Table 4.** A summary of mean bias (MB) and root mean square error (RMSE) for parametric and nonparametric methods in stand development. Data collected from the long-term experimental plots installed in the National Taiwan University Experimental Forest (NTUEF) were used as a fitting dataset, while those collected from the National Forest Resource Inventory (NFRI) permanent plots were utilized for model validation.

| Fitting Data | | | | | | | | |
|---|---|---|---|---|---|---|---|---|
| Method | Stage I | | Phase I | | Stage II Phase II | | Phase III | |
| | MB | RMSE | MB | RMSE | MB | RMSE | MB | RMSE |
| Segmented Regression (Full) | −0.002 | 0.059 | 0.000 | 0.032 | −0.058 | 0.060 | 0.002 | 0.105 |
| Segmented Regression (Reduced) | −0.061 | 0.073 | 0.080 | 0.039 | −0.032 | 0.062 | −0.016 | 0.106 |
| Penalized Spline | −0.004 | 0.054 | 0.005 | 0.032 | −0.019 | 0.060 | 0.004 | 0.098 |
| Random Forest for Regression | 0.010 | 0.021 | −0.003 | 0.018 | 0.002 | 0.032 | −0.002 | 0.052 |
| Validation Data | | | | | | | | |
| Method | Stage I | | Phase I | | Stage II Phase II | | Phase III | |
| | MB | RMSE | MB | RMSE | MB | RMSE | MB | RMSE |
| Segmented Regression (Full) | - | - | −0.015 | 0.141 | −0.058 | 0.158 | 0.179 | 0.154 |
| Segmented Regression (Reduced) | - | - | 0.049 | 0.144 | −0.033 | 0.158 | 0.143 | 0.150 |
| Penalized Spline | - | - | −0.013 | 0.141 | −0.019 | 0.159 | 0.192 | 0.164 |
| Random Forest for Regression | - | - | −0.003 | 0.140 | −0.021 | 0.175 | 0.202 | 0.212 |

Indeed, the main advantage of using segmented regression models is to provide detailed information on stand development, such as the timing and magnitude of self-thinning. However, identifying proper initial values or shape of the segments is difficult and challenging. With an increasing number of parameters, convergence is hardly obtained with poor initial starting values. As indicated by VanderSchaaf and Burkhart [3], the convergence criteria were not always met in parameter estimation when the full segmented regression model was fit to the data among various planting densities for loblolly pine. In contrast, the nonparametric methods, like penalized spline or random forest for regression approaches, are advantageous because they do not require to specify the initial starting values in parameter estimation nor rely on strong distribution assumptions and model specification. However, in order to implement penalized spline or random forest algorithms, a number of choices on hyperparameters (tuning parameters) need to be made, such as number of knots, smoothing parameter, number of decision trees.

Horowitz [31] pointed out that nonparametric methods could be limited when predicting what might happen under conditions that do not exist in the available data. As mentioned above, the random forest algorithm made slightly more accurate predictions of the size–density relationship for the training data than those for the validation data. The difference of predictability may be due to the nature of the ensemble algorithm, where the prediction is made by averaging the results from a large number of decision trees in the training data. Thus, the inherent inability makes predictions beyond the range of the original training data inaccurate (i.e., poor extrapolation) [32]. In this study, although an independent dataset was used for model evaluation, the range of the validation dataset is within the range of the fitting data (see Table 1). Therefore, the random forest algorithm can still be used to predict stand development well in this study. In this study, two independent datasets were used as fitting and validation data, while other data splitting schemes, such as random selection of a small portion of data for validation, will be worth to investigate for future studies. From a statistical point of view, extrapolation from the original range may produce poor estimates but in forestry practice, the extrapolation could be needed when implementing the model due to the time and cost constraints. Thus, the robustness of the method should be considered when random forest or other ensemble algorithms (e.g., bagging trees) are applied. Notably, random forest algorithm can also be used in variable selection. To be comparable with other models, only two predictors were used in this study. Adding additional predictors, such as environmental and climatic factors, is suggested to explore the applicability of this algorithm.

Various spline regression approaches have proved highly flexible and robust [33]. Spline-based regression approaches have been widely applied and examined in tree stem taper estimation [16,34]. In this study, we found that penalized spline can also be used to predict size–density relationships at various stages in the long-term stand development. In addition, Hazelton et al. [33] indicated that the nonparametric regression method, like splines, can be used as an important tool for data exploration and visualization in the preliminary study before conducting formal model fitting.

Generally, Japanese cedar is harvested at 30-40 years. As shown in Figure 2, although an obvious decline in the number of trees was observed in some stands, many of them still stayed along the self-thinning line when the quadratic mean diameter exceeded 40.4 cm ($LnD_q = 3.7$). It implied that Japanese cedar in Taiwan has potential to produce large size logs. In addition to timber production, Japanese cedar plantations serve as important wildlife habitats. The long-term stand development can be accurately described by the parametric and nonparametric methods used in this study, which will provide useful information on scheduling silvicultural treatments and management practices for old-growth Japanese cedar forests. According to the current results, the penalized spline approach is recommended for estimating the long-term size–density relationships for Japanese cedar because of its accuracy, robustness, and straightforward implementation. Segmented regression models can be useful when comparing the self-thinning behaviors with other studies. However, parameter estimation may be challenging if inappropriate initial values are chosen. It should be noted that we didn't examine the interspecific competition, which is a limitation of the models used in this study. When multiple species

are vigorous in the stand, quantifying the impact of various competitors is important. Compared to Reineke's SDI, Yoda's self-thinning rule has been explicitly investigated for Japanese cedar [20,21]. In this study, the models used described the $LnN - LnD_q$ relationships instead of the relationships between tree volume and number of trees per unit area, which provided a different perspective on self-thinning behaviors. Indeed, Reineke's stand density index and Yoda's self-thinning rule are mathematically equivalent with the assumption of the allometric relationship between tree volume and quadratic mean diameter [1]. Other self-thinning indices, such as competition–density index, are also related to Yoda's self-thinning rule [9]. Future studies on incorporating other self-thinning indices into the size–density trajectory models are warranted.

## 4. Conclusions

In summary, segmented regression models, penalized spline and random forest for regression methods can provide reliable estimates of the size–density relationships for long-term Japanese cedar plantations in Taiwan. Higher accuracy was achieved before the stands diverged from the self-thinning line. The segmented regression models proposed by VanderSchaaf and Burkhart [3] are useful to describe the entire stand development trajectories, while identifying proper initial starting values and shape of segments can be challenging and difficult. The penalized spline approach behaved consistently well regardless of datasets or stages in stand development, while the predictability of the random forest algorithm slightly decreased when the validation data was fitted. The penalized spline approach is recommended because of its accuracy and robustness. Additionally, the method is easy to be implemented.

In this study, the tree mortality induced by intraspecific competition occurred when quadratic mean diameter approached between 6.25 and 10.9 cm ($LnD_q = 1.83 - 2.39$). The slope of self-thinning line estimated is steeper than in northeastern Japan. In addition, although an obvious decline of the number of trees was observed in some stands, it was found that many of the stands still stayed along the self-thinning line, which implied that Japanese cedar in Taiwan has potential to produce large size timber. In this study, we focused on the long-term relationships between quadratic mean diameter and number of trees per unit area. Future studies on incorporating other self-thinning indices or adding environmental and climatic variables into the size–density trajectory models are warranted.

**Author Contributions:** Conceptualization, C.-P.C. and S.-I.Y.; Methodology, S.-I.Y.; Software, S.-I.Y.; Validation, S.-I.Y.; Formal Analysis, C.-P.C. and S.-I.Y.; Investigation, C.-P.C.; Resources, C.-R.C.; Data Curation, C.-R.C. and C.-P.C.; Writing—Original Draft Preparation, C.-P.C. and S.-I.Y.; Writing—Review & Editing, C.-R.C., C.-P.C., S.-I.Y.; Visualization, C.-P.C.; Supervision, C.-R.C.; Project Administration, C.-P.C.; Funding Acquisition, C.-R.C. All authors have read and agreed to the published version of the manuscript.

**Funding:** This research received no external funding.

**Acknowledgments:** We gratefully acknowledge Taiwan Forestry Bureau and National Taiwan University Experimental Forest for providing Japanese cedar long-term data in Taiwan. Comments from three anonymous reviewers are gratefully appreciated. This article was subsidized by National Taiwan University (NTU), Taiwan.

**Conflicts of Interest:** The authors declare no conflict of interest.

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
