# Peer review of "Comparison of Parametric and Nonparametric Methods for Estimating Size–Density Relationships in Old-Growth Japanese Cedar (Cryptomeria japonica D. Don) Plantations"

_forests, doi:10.3390/f11060625_

Round 1

Reviewer 1 Report

The overall quality of the paper is good. Data and methods are correctly presented and shows good scientific soundness. Results are clearly shown, although some more discussion would be appreciated. The topic (size-density relationship) has been widely studied in the literature, but a analyzing other statistical procedures to approach the issue is interesting.

There is some lack of discussion, leading to have conclusions that have already be shown in results. Sometimes, just copying the results.  

Lines 268-270 should go to discussion/conclusions.

Some discussion would be desirable, for instance, regarding suggestions on which method outstands, even if a little. Maybe some contribution whether differences among full and reduced segmented models can be used for decision making, if parsimony plays any role here...

The true statement of the usual poorer behaviour of Random Forest over validation datasets could just be used also to make some discussion, since it looks like the results are not as good as others.

This added discussion would enrich the overall quality of the paper

Author Response

All line numbers in the response refer to the revised submission.

Point 1.1: The overall quality of the paper is good. Data and methods are correctly presented and shows good scientific soundness. Results are clearly shown, although some more discussion would be appreciated. The topic (size-density relationship) has been widely studied in the literature, but analysing other statistical procedures to approach the issue is interesting.

There is some lack of discussion, leading to have conclusions that have already be shown in results. Sometimes, just copying the results.

(Response 1.1) Thank you for the comments. More information has been added in the discussion section as suggested. The conclusion section has been revised.

Point 1.2: Lines 268-270 should go to discussion/conclusions.

(Response 1.2) Thanks for the suggestion. Sentences were added in the conclusions (see Lines 344-346). 

Point 1.3: Some discussion would be desirable, for instance, regarding suggestions on which method outstands, even if a little. Maybe some contribution whether differences among full and reduced segmented models can be used for decision making, if parsimony plays any role here...

(Response 1.3) Appreciate the comments! More discussion has been added as suggested (see Lines 235-243).

Point 1.4: The true statement of the usual poorer behaviour of Random Forest over validation datasets could just be used also to make some discussion, since it looks like the results are not as good as others. This added discussion would enrich the overall quality of the paper

(Response 1.4) Thanks for pointing out. Suggestions of using the random forest algorithm were given in Lines 291-298. 

Reviewer 2 Report

Manuscript ID: forests-789226

Title: Comparison of parametric and nonparametric methods for estimating size-density relationships in old-growth Japanese cedar plantations

Overall comments

As this article focuses on the size-density relationships in Japanese cedar planation of Taiwan, based on the scope of this journal, it is suggested for publication in this journal, Forests. As the author indicated in their manuscript, this kind of study has not been addressed commonly yet. Thus, the originality of this work lies in the comparison of parametric and nonparametric methods for size-density relationships especially in Japanese cedar plantation of Taiwan. This article would be of interest to forest modelers.

This manuscript was judged to be well-organized with clear research objectives and methodologies. Also, the data sources used in this study should be adequate to evaluate the model. The manuscript was generally readable with concise paragraphs and sentences. The author did compare the parametric and nonparametric methods. However, it may be improved by mentioning the practical point.

There are several parts which can be improved by reconsidering and revising it. It is expected to be more concrete and logical after revision. Specific comments are as follows.

Title

It is well selected.

Abstract

Your abstract concisely summarized your main results and briefly mentioned some of the techniques you used to arrive at them.

There is no comment for this section.

keywords

I fully agree the keywords here.

Introduction

The introduction provided adequate historical context and scientific rationale for this research. The research objectives were well supported by this logic.

Line 59: However, if Japanese cedar is an important species, as referred by the authors, wasn’t there any study on size-density relationship for the species?

If it exists, what would be the difference from the previous one? Or can you compare this study to Nishizono and Tanaka (2012)? I wonder about the features which can solve the current problem.

It is needed to emphasize the necessity from the practical point of view. (even though the topic is interesting, will it be helpful for Japanese cedar to be managed?)

Materials and Methods

2.1.1. National Forest Resource Inventory (NFRI) permanent plots

Line 72: the plot size of 0.02 ha seems to be small.

Assume that a stand has dbh 55.6cm, tph 340, age 93 according to the table 1, there might be less than 7 trees in plot of 0.02 ha. Will the number of sample tree be represented for a stand sample?

Or there is no such a case in this data? It is better to explain about this.

Line 75: Wasn’t there any difference of silviculture practice among the offices? Or Did author just not consider this kind of effect even though the silviculture practice was quite different?

It should be described more to explain the controlled variables

Line 80: A criterion, plots with Japanese cedar basal area greater 30%, is quite low.

How do we say this is a plantation? Are the authors sure that the censored sample plots are Japanese cedar plantation?

If the other species are vigorous, it probably influences the size-density relationship a lot.

Is there any other reference for this criterion, BA > 30%?

In my opinion, it should be greater than 50% at least, especially because the data is originated from NFI.

 2.1.2. Long-term experimental plots

Line 98: Add the variable Dq, quadratic mean diameter, in table 1 which was used as a predictor.

2.2.1. Segmented regression models

Methods section was also systematic and generally described the procedures well.

However, the detail may not be enough even though the author cited accurate replication by experienced modelers, e.g. VanderSchaaf and Burkhart (2008).

The method section should describe more about the model you selected.

Line 120: What is the meaning of Reduced model? Why did you compare Reduced model with Full model?

What information can readers get? What is this for? Application, biometric concept, or accurate estimation?

Specify more about this. The explanation is not enough in the manuscript.

2.2.3. Random forest for regression

As described in the manuscript, Random forest takes variables at random for each of decision trees. However, there were only two independent variables considered in this study.

Two independent variables are quite limited for random selection in random forest ensemble method. How about just decision tree and/or bagging?

In addition, how did you tune the hyperparameters such as max. depth, min. samples leaf, and/or min. sample split as well as no. of decision trees?

The results were overfitted probably because of this.

2.3. Method evaluation

Line 147-149: If the method of splitting data was not applied only for random forest, but also for Segmented regression models and Penalized spline, then it should be explained in the section of 2.3. Method evaluation

Results and discussion

Your Results section was well organized and objectively presented your data and analysis.

However, it can be improved with the support of additional graph information.

Line 168, Figure 2: The predicted line is short. Draw it a little bit longer so that all the data samples are estimated within the line, especially in Fig. 2a.

3.1. Size-density relationships estimated by the segmented regression models

Definite comparison is not enough between two segmented regressions except for mentioning the parameter estimates.

Which one is better? Which one can the author recommend?

Your discussion offered reasonable and logical explanations for your discoveries. In addition, many of points were upheld by pertinent citations.

Line 217-223: I agreed to the author’s discussion. It may be because of the overfitting of random forest.

Line 249-258: I agreed to the author’s discussion. To solve this problem, why didn’t you mix the two dataset and shuffle it for training and test data?

line 266-268: It is better to insert a figure 3 with arithmetic N vs. arithmetic Dq, if the authors try to make readers understandable for actual self-thinning in dataset. 

line 273-276, 293-294: In my opinion, authors do not need to mention this kind of exceptional situation. I don’t think much this is the limitation of the present study. The manuscript is consistent and logical enough even without these sentences. Rather than mentioning these, I suggest the authors discuss the optimal model, pros and cons, based on the situation.

  1. Conclusions

It is organized well. No comments on this section.

Additional comments on data split

Why did the authors separate training and test dataset in this way?

Would it be better to combine all together, and then split it as training and testing data?

Was there any specific purpose? For instance, was this model for Long-term experimental plots than NFRI permanent plots?

If there is any additional information which can help readers understand, explain it in method section.

Author Response

All line numbers in the response refer to the revised submission.

  1. In response to the reviewer’s overall comments

Point 1.1: As this article focuses on the size-density relationships in Japanese cedar planation of Taiwan, based on the scope of this journal, it is suggested for publication in this journal, Forests. As the author indicated in their manuscript, this kind of study has not been addressed commonly yet. Thus, the originality of this work lies in the comparison of parametric and nonparametric methods for size-density relationships especially in Japanese cedar plantation of Taiwan. This article would be of interest to forest modelers.

This manuscript was judged to be well-organized with clear research objectives and methodologies. Also, the data sources used in this study should be adequate to evaluate the model. The manuscript was generally readable with concise paragraphs and sentences. The author did compare the parametric and nonparametric methods. However, it may be improved by mentioning the practical point.There are several parts which can be improved by reconsidering and revising it. It is expected to be more concrete and logical after revision.

(Response 1.1) We greatly appreciate your effort and invaluable comments on this manuscript. We have carefully addressed all issues indicated in the review report and believe that the revision is suitable for further consideration for publication.

  1. In response to the reviewer’s suggestions in the Introduction

Point 2.1: The introduction provided adequate historical context and scientific rationale for this research. The research objectives were well supported by this logic. Line 59: However, if Japanese cedar is an important species, as referred by the authors, wasn’t there any study on size-density relationship for the species? If it exists, what would be the difference from the previous one? Or can you compare this study to Nishizono and Tanaka (2012)? I wonder about the features which can solve the current problem. It is needed to emphasize the necessity from the practical point of view. (even though the topic is interesting, will it be helpful for Japanese cedar to be managed?)

(Response 2.1) Thanks for the suggestion. More justification and clarification of this study were added in the Introduction. The practical implication of this study was also emphasized (see Lines 64-86).

  • In response to the reviewer’s suggestions in the Materials and Methods

Point 3.1 2.1.1. National Forest Resource Inventory (NFRI) permanent plots. Line 72: the plot size of 0.02 ha seems to be small. Assume that a stand has dbh 55.6cm, tph 340, age 93 according to the table 1, there might be less than 7 trees in plot of 0.02 ha. Will the number of sample tree be represented for a stand sample? Or there is no such a case in this data? It is better to explain about this.

(Response 3.1) Thanks for the comment. We agree that the plot size was not as large as some ecological long-term study plots, while the plot size is commonly used in forestry practice. The total number of sample trees ranged from 10 to 88 trees per plot at stand ages of 8-93 (yrs). (see Lines 105)

Point 3.2 Line 75: Wasn’t there any difference of silviculture practice among the offices? Or Did author just not consider this kind of effect even though the silviculture practice was quite different? It should be described more to explain the controlled variables

(Response 3.2) Each office followed the identical standard of the silviculture practice regardless of sites or elevations. In addition to herbaceous vegetation removal and seedling replanting, no other treatments were applied on the permanent plots. The details have been added (see Lines 94-97)

Point 3.3 Line 80: A criterion, plots with Japanese cedar basal area greater 30%, is quite low. How do we say this is a plantation? Are the authors sure that the censored sample plots are Japanese cedar plantation? If the other species are vigorous, it probably influences the size-density relationship a lot. Is there any other reference for this criterion, BA > 30%? In my opinion, it should be greater than 50% at least, especially because the data is originated from NFI.

(Response 3.3) Thanks for pointing out. More details on plot selection have been added (see Lines 101-105). Plots with a majority of poor-quality standing trees or dominated by other weedy species were not included in analyses. A total number of 515 observations were collected from 222 permanent plots where Japanese cedar basal area was greater than 30% of the total basal area and Japanese cedar dominated at least 20% of the main canopy. Among all observations, the total number of sample trees ranged from 10 to 88 trees per plot.

Point 3.4:  2.1.2. Long-term experimental plots. Line 98: Add the variable Dq, quadratic mean diameter, in table 1 which was used as a predictor.

(Response 3.4) Thanks for pointing out. Done. The variable Dq, quadratic mean diameter has been added.

Point 3.5: 2.2.1. Methods section was also systematic and generally described the procedures well. However, the detail may not be enough even though the author cited accurate replication by experienced modelers, e.g. VanderSchaaf and Burkhart (2008). The method section should describe more about the model you selected.

Line 120: What is the meaning of Reduced model? Why did you compare Reduced model with Full model? What information can readers get? What is this for? Application, biometric concept, or accurate estimation? Specify more about this. The explanation is not enough in the manuscript.

(Response 3.5) More descriptions of the full model and reduced model were added in this section (see Lines 143-149).

Point 3.6: 2.2.3. Random forest for regression. As described in the manuscript, Random forest takes variables at random for each of decision trees. However, there were only two independent variables considered in this study. Two independent variables are quite limited for random selection in random forest ensemble method. How about just decision tree and/or bagging? In addition, how did you tune the hyperparameters such as max. depth, min. samples leaf, and/or min. sample split as well as no. of decision trees? The results were overfitted probably because of this.

(Response 3.6) Thanks for the comments. The description of the selection of hyperparameters was added (see Lines 173-174). We agree that the random forest algorithm may not be powerful with the small number of predictors. However, since the random forest algorithm is the most popular method compared to other ensemble algorithms, the results of this study can add some additional insights of using this method. In the results and discussion section, more discussion has been added (see Lines 283-291).     

Point 3.7: 2.3. Method evaluation. Line 147-149: If the method of splitting data was not applied only for random forest, but also for Segmented regression models and Penalized spline, then it should be explained in the section of 2.3. Method evaluation

(Response 3.7) Clarification made (see Lines 176-188).   

  1. In response to the reviewer’s suggestion in the Results and Discussion

Point 4.1: Your Results section was well organized and objectively presented your data and analysis. However, it can be improved with the support of additional graph information. Line 168, Figure 2: The predicted line is short. Draw it a little bit longer so that all the data samples are estimated within the line, especially in Fig. 2a.

(Response 4.1) Thanks for catching this. The predicted lines were modified to cover the entire range of the data.

Point 4.2: 3.1. Size-density relationships estimated by the segmented regression models. Definite comparison is not enough between two segmented regressions except for mentioning the parameter estimates. Which one is better? Which one can the authors recommend? Your discussion offered reasonable and logical explanations for your discoveries. In addition, many of points were upheld by pertinent citations.

(Response 4.2) Appreciate the comments! More discussion and suggestions were added (see Lines 235-243).

Point 4.3: 3.1. Size-density relationships estimated by the segmented regression models Line 217-223: I agreed to the author’s discussion. It may be because of the overfitting of random forest.

Line 249-258: I agreed to the author’s discussion. To solve this problem, why didn’t you mix the two dataset and shuffle it for training and test data?

(Response 4.3) Thanks for the suggestion. We agree that it is common to randomly select a small portion of the data as a validation dataset. However, in this study, we think it is more appropriate to keep the data separated because the data were obtained from two different sources. The clarification has been given in Lines 176-185. 

The two fully independent datasets can also be helpful to evaluate the robustness of the methods/models. We understand that from a statistical point of view, extrapolation from the original range can produce poor estimates but in forestry practice, the extrapolation could sometimes be required when implementing the models/methods due to the time and cost constraints. The discussion has been added as well (see Lines 285-291).

Point 4.4: line 266-268: It is better to insert a figure 3 with arithmetic N vs. arithmetic Dq, if the authors try to make readers understandable for actual self-thinning in dataset.

(Response 4.4) Thanks for pointing out. We rephrased the sentence to be more understandable.

Point 4.5: line 273-276, 293-294: In my opinion, authors do not need to mention this kind of exceptional situation. I don’t think much this is the limitation of the present study. The manuscript is consistent and logical enough even without these sentences. Rather than mentioning these, I suggest the authors discuss the optimal model, pros and cons, based on the situation.

(Response 4.5)  Thanks for the suggestion. We agree. The sentences have been replaced as suggested.

  1. In response to the reviewer’s additional comments

Point 5.1: Why did the authors separate training and test dataset in this way? Would it be better to combine all together, and then split it as training and testing data? Was there any specific purpose? For instance, was this model for Long-term experimental plots than NFRI permanent plots? If there is any additional information which can help readers understand, explain it in method section.

(Response 5.1) We appreciate your suggestions and comments on data splitting. Additional clarifications have been added in the materials and methods section (see Lines 176-185). As mentioned above, we agree that random selection of a subset from the combined dataset is a commonly-used data splitting approach. However, because the data were collected from two different sources, we think it is more appropriate to keep the data separated. The two fully independent datasets were also advantageous to evaluate the robustness of the methods/models. Although the number of the NTUEF long-term experimental plots is less than that of the NFRI permanent plots, more remeasurements per plot were taken. Thus, the NTUEF plots were more suitable in model/method fitting/training to capture the complete size-density relationship trajectories.

Reviewer 3 Report

Authors have compared parametric and nonparametric methods for estimating size-density relationship for Japanese cedar plantations in Taiwan. They also compared the efficacy of two segmented regression models with the penalized spline and random forest for regression methods. This research directly related to self thinning model and surely can be used for forest management and silviculture practices.  However, my main concern is why authors did not checked other self thinning models such as Akio Hagihara (2014) Deriving the mean mass-density trajectory by reconciling the competition-density effect law with the self-thinning law in even-aged pure stands, Journal of Forest Research, 19:1, 125-133, DOI: 10.1007/s10310-013-0393-2? My major concern is to also check other models and compare them with nonparametric and parametric models to understand which model fits well. 

Introduction need to be improved and add one paragraph about self-thinning. Origin of self thining model and how it progressed over time in terms of model development and later can be discussed about which model is good and would provide better results. 

Below are some of my minor comments

Title – Change Japanese cedar to its scientific name
Key words – Delete scientific name
Line 34 – Change per unit acre to per unit area
Line 89 – How does plot size and elevation change would affect size-density curve? Need to discuss in discussion part. Also can add in conclusion that this could be future research which need to be carry out in future.
Table or text – number of trees per hectare (TPH) – change TPH to rho
In method section add other models.
Change table 3 to table 2 and vice versa.
Figure 2b – comparison of two segmented regression – two is missing (also add smaller ticks in the figure with quadrates in x and y axis to see the self-thinning point clearly.
Line 211 – I do not see these values in table 3.
Line 244 – Need to add figures for penalized spline or random forest models. How data looks likes in figure after fitting?
Line 270-273- Authors conclusion is that both models can be used for forest management and silviculture practices but need to describe more that which one is the best based on output.

Author Response

  1. In response to the reviewer’s general comments

Point 1.1: Authors have compared parametric and nonparametric methods for estimating size-density relationship for Japanese cedar plantations in Taiwan. They also compared the efficacy of two segmented regression models with the penalized spline and random forest for regression methods. This research directly related to self thinning model and surely can be used for forest management and silviculture practices.  However, my main concern is why authors did not checked other self thinning models such as Akio Hagihara (2014) Deriving the mean mass-density trajectory by reconciling the competition-density effect law with the self-thinning law in even-aged pure stands, Journal of Forest Research, 19:1, 125-133, DOI: 10.1007/s10310-013-0393-2? My major concern is to also check other models and compare them with nonparametric and parametric models to understand which model fits well.

(Response 1.1) Thank you very much for providing the invaluable comments and useful references. We fully agree that many self-thinning models/methods were developed in the past. Comparison of the self-thinning models/methods has been widely discussed in forestry literature. The relevant studies have been added (see Lines 44-86). To our knowledge, fewer studies focused on the modelling the stand development outside the self-thinning stage. Therefore, this study aimed at estimating the full trajectory of the size-density relationships, especially for the old-growth forests. The objective of this study was to compare the pros and cons of different models/methods when describing the size-density relationships, which can serve as some guidelines for forest managers in practice. We appreciate the suggestions. Incorporating various self-thinning models into the size-density relationship models are warranted, which was suggested for the future studies (see Lines 355-357).  

Point 1.2: Introduction need to be improved and add one paragraph about self-thinning. Origin of self-thinning model and how it progressed over time in terms of model development and later can be discussed about which model is good and would provide better results. 

(Response 1.2) Thanks for the comments. More descriptions of self-thinning models with relevant references have been added in the Introduction (see Lines 44-86).

  1. In response to the reviewer’s minor comments

Point 2.1: Title – Change Japanese cedar to its scientific name. Keywords – Delete scientific name

(Response 2.1) Both common name and scientific name have been added in the title. The scientific name in the Keywords was removed.

Point 2.2: Line 34 – Change per unit acre to per unit area

(Response 2.2) Appreciate the comments! We change per unit acre to per unit area

Point 2.3: Line 89 – How does plot size and elevation change would affect size-density curve? Need to discuss in discussion part. Also, can add in conclusion that this could be future research which need to be carry out in future.

(Response 2.3) Thanks for pointing out. The site scale of this study is on the whole of Taiwan Island. In the case of site scale, more attention will be paid to site micro-environment factors such as temperature, humidity, slope, aspect and soil condition at site scale in future studies.  This is also one of the interesting topic for future research (see Line355-357). More details on plot size (see Lines 90-93) and discussion part (see Lines 206-220) have been added. Elevation change affects temperature, current observations show that low temperature is one of the growth of factors, still needs to be confirmed in Taiwan Island.

Point 2.4: Table or text – number of trees per hectare (TPH) – change TPH to rho 

(Response 2.4) Thanks for the suggestion on the abbreviation. We think TPH (trees per ha) is commonly used in forestry literature.

Point 2.5: In method section add other models. 

(Response 2.5) Thanks for the suggestion. In the future, we will arm to combine and compare different self-thinning models or other analytical methodologies, and submit a manuscript to study this issue (see Line355-357). The main purpose of the article is to focus on a new analysis method to fit the different stages of the size-density relationship, to Comparing the difference between parametric and nonparametric methods.

Point 2.6: Change table 3 to table 2 and vice versa

(Response 2.6) Thanks for the suggestion. We think the original order of the tables is fine.

Point 2.7: Figure 2b – comparison of two segmented regression – two is missing (also add smaller ticks in the figure with quadrates in x and y axis to see the self-thinning point clearly.

(Response 2.7) Thanks for the comment. Graphs modified.

Point 2.8: Line 211 – I do not see these values in table 3

 (Response 2.8) Thanks for pointing out. The sentence has been re-written.

Point 2.9: Line 244 – Need to add figures for penalized spline or random forest models. How data looks likes in figure after fitting? 

 (Response 2.9) Thanks for the suggestion. The performance of the four methods has been given in Tables 3 and 4. We think it is easier to check the difference among four methods using tables than figures.

Point 2.10: Line 270-273- Authors conclusion is that both models can be used for forest management and silviculture practices but need to describe more that which one is the best based on output.

 (Response 2.10) Thanks for the comments. More descriptions of model with the best based on output have been added in the 3.2. (see Line 256-269) Compared to other three methods used, the penalized spline approach behaved consistently well regardless of datasets or stand development stages. Penalized spline approach is easier to be implemented as well. Segmented regression models can be useful when comparing the self-thinning behaviors with other studies. (see Line 316-319)

Round 2

Reviewer 2 Report

Journal: Forests

Manuscript ID: forests-789226

Title: Comparison of parametric and nonparametric methods for estimating size-density relationships in old-growth Japanese cedar plantations

Reviewer 2

In this second rounding of peer-review, my comments are blue color in the attached Word file.

  1. In response to the reviewer’s overall comments

Point 1.1: As this article focuses on the size-density relationships in Japanese cedar planation of Taiwan, based on the scope of this journal, it is suggested for publication in this journal, Forests. As the author indicated in their manuscript, this kind of study has not been addressed commonly yet. Thus, the originality of this work lies in the comparison of parametric and nonparametric methods for size-density relationships especially in Japanese cedar plantation of Taiwan. This article would be of interest to forest modelers.

This manuscript was judged to be well-organized with clear research objectives and methodologies. Also, the data sources used in this study should be adequate to evaluate the model. The manuscript was generally readable with concise paragraphs and sentences. The author did compare the parametric and nonparametric methods. However, it may be improved by mentioning the practical point. There are several parts which can be improved by reconsidering and revising it. It is expected to be more concrete and logical after revision.

(Response 1.1) We greatly appreciate your effort and invaluable comments on this manuscript. We have carefully addressed all issues indicated in the review report and believe that the revision is suitable for further consideration for publication.

è Thank you for authors’ effort. I could feel the authors have carefully revised this manuscript. It was highly improved. Attached are my additional comments.

  1. In response to the reviewer’s suggestions in the Introduction

Point 2.1: The introduction provided adequate historical context and scientific rationale for this research. The research objectives were well supported by this logic. Line 59: However, if Japanese cedar is an important species, as referred by the authors, wasn’t there any study on size-density relationship for the species? If it exists, what would be the difference from the previous one? Or can you compare this study to Nishizono and Tanaka (2012)? I wonder about the features which can solve the current problem. It is needed to emphasize the necessity from the practical point of view. (even though the topic is interesting, will it be helpful for Japanese cedar to be managed?)

(Response 2.1) Thanks for the suggestion. More justification and clarification of this study were added in the Introduction. The practical implication of this study was also emphasized (see Lines 64-86).

            è Now it’s clear. No more comments on this issue.

III. In response to the reviewer’s suggestions in the Materials and Methods

Point 3.1 2.1.1. National Forest Resource Inventory (NFRI) permanent plots. Line 72: the plot size of 0.02 ha seems to be small. Assume that a stand has dbh 55.6cm, tph 340, age 93 according to the table 1, there might be less than 7 trees in plot of 0.02 ha. Will the number of sample tree be represented for a stand sample? Or there is no such a case in this data? It is better to explain about this.
(Response 3.1) Thanks for the comment. We agree that the plot size was not as large as some ecological long-term study plots, while the plot size is commonly used in forestry practice. The total number of sample trees ranged from 10 to 88 trees per plot at stand ages of 8-93 (yrs). (see Lines 105)

            è I partially agree that 0.02 ha plot size is used in forestry practice. However, it would be rather suitable for young stands. It should not use for old and large stands. Fortunately, there were 10 trees per plot at least. Still, it is relatively small, but understandable.
The number of sample trees per plot is important for many statistical approaches. Thus, I wish the authors should be careful with this matter in future work.
The authors’ explanation was clear. no more comments on this issue.

Point 3.2 Line 75: Wasn’t there any difference of silviculture practice among the offices? Or Did author just not consider this kind of effect even though the silviculture practice was quite different? It should be described more to explain the controlled variables.

(Response 3.2) Each office followed the identical standard of the silviculture practice regardless of sites or elevations. In addition to herbaceous vegetation removal and seedling replanting, no other treatments were applied on the permanent plots. The details have been added (see Lines 94-97)

            è The authors’ explanation was informative. It is a clear description. no more comments on this issue.

Point 3.3 Line 80: A criterion, plots with Japanese cedar basal area greater 30%, is quite low. How do we say this is a plantation? Are the authors sure that the censored sample plots are Japanese cedar plantation? If the other species are vigorous, it probably influences the size-density relationship a lot. Is there any other reference for this criterion, BA > 30%? In my opinion, it should be greater than 50% at least, especially because the data is originated from NFI.

(Response 3.3) Thanks for pointing out. More details on plot selection have been added (see Lines 101-105). Plots with a majority of poor-quality standing trees or dominated by other weedy species were not included in analyses. A total number of 515 observations were collected from 222 permanent plots where Japanese cedar basal area was greater than 30% of the total basal area and Japanese cedar dominated at least 20% of the main canopy. Among all observations, the total number of sample trees ranged from 10 to 88 trees per plot.

            è Ok. It is clearer. However, I suggest the authors discuss the model self-evaluation with this criterion. This factor can be dealt with one of the model limits.

Point 3.4: 2.1.2. Long-term experimental plots. Line 98: Add the variable Dq, quadratic mean diameter, in table 1 which was used as a predictor.

(Response 3.4) Thanks for pointing out. Done. The variable Dq, quadratic mean diameter has been added.

è Ok. It is better. no more comments on this issue.

Point 3.5: 2.2.1. Methods section was also systematic and generally described the procedures well. However, the detail may not be enough even though the author cited accurate replication by experienced modelers, e.g. VanderSchaaf and Burkhart (2008). The method section should describe more about the model you selected. Line 120: What is the meaning of Reduced model? Why did you compare Reduced model with Full model? What information can readers get? What is this for? Application, biometric concept, or accurate estimation? Specify more about this. The explanation is not enough in the manuscript

(Response 3.5) More descriptions of the full model and reduced model were added in this section (see Lines 143-149).

è It is better. Thank you for authors’ effort. no more comments on this issue.

Point 3.6: 2.2.3. Random forest for regression. As described in the manuscript, Random forest takes variables at random for each of decision trees. However, there were only two independent variables considered in this study. Two independent variables are quite limited for random selection in random forest ensemble method. How about just decision tree and/or bagging? In addition, how did you tune the hyperparameters such as max. depth, min. samples leaf, and/or min. sample split as well as no. of decision trees? The results were overfitted probably because of this.

(Response 3.6) Thanks for the comments. The description of the selection of hyperparameters was added (see Lines 173-174). We agree that the random forest algorithm may not be powerful with the small number of predictors. However, since the random forest algorithm is the most popular method compared to other ensemble algorithms, the results of this study can add some additional insights of using this method. In the results and discussion section, more discussion has been added (see Lines 283-291).

è It is better. no more comments on this issue.

Point 3.7: 2.3. Method evaluation. Line 147-149: If the method of splitting data was not applied only for random forest, but also for Segmented regression models and Penalized spline, then it should be explained in the section of 2.3. Method evaluation

(Response 3.7) Clarification made (see Lines 176-188).

è It should be as it is. Now it is clear. no more comments on this issue.

  1. In response to the reviewer’s suggestion in the Results and Discussion

Point 4.1: Your Results section was well organized and objectively presented your data and analysis. However, it can be improved with the support of additional graph information. Line 168, Figure 2: The predicted line is short. Draw it a little bit longer so that all the data samples are estimated within the line, especially in Fig. 2a.

(Response 4.1) Thanks for catching this. The predicted lines were modified to cover the entire range of the data.

è Now it is clearer. no more comments on this issue.

Point 4.2: 3.1. Size-density relationships estimated by the segmented regression models. Definite comparison is not enough between two segmented regressions except for mentioning the parameter estimates. Which one is better? Which one can the authors recommend? Your discussion offered reasonable and logical explanations for your discoveries. In addition, many of points were upheld by pertinent citations.

(Response 4.2) Appreciate the comments! More discussion and suggestions were added (see Lines 235-243).

è Great. Now it is clearer. no more comments on this issue.

Point 4.3: 3.1. Size-density relationships estimated by the segmented regression models Line 217-223: I agreed to the author’s discussion. It may be because of the overfitting of random forest. Line 249-258: I agreed to the author’s discussion. To solve this problem, why didn’t you mix the two dataset and shuffle it for training and test data?

(Response 4.3) Thanks for the suggestion. We agree that it is common to randomly select a small portion of the data as a validation dataset. However, in this study, we think it is more appropriate to keep the data separated because the data were obtained from two different sources. The clarification has been given in Lines 176-185.

The two fully independent datasets can also be helpful to evaluate the robustness of the methods/models. We understand that from a statistical point of view, extrapolation from the original range can produce poor estimates but in forestry practice, the extrapolation could sometimes be required when implementing the models/methods due to the time and cost constraints. The discussion has been added as well (see Lines 285-291).

è Ok. I understand the data characteristics. I suggest the authors note this issue in Discussion section. With the authors’ comment above, it is more logical and concrete if the authors write additional short comments about the caution of interpreting model fitting and validation.

Point 4.4: line 266-268: It is better to insert a figure 3 with arithmetic N vs. arithmetic Dq, if the authors try to make readers understandable for actual self-thinning in dataset.

(Response 4.4) Thanks for pointing out. We rephrased the sentence to be more understandable.

è Ok. No more comment on this issue.

Point 4.5: line 273-276, 293-294: In my opinion, authors do not need to mention this kind of exceptional situation. I don’t think much this is the limitation of the present study. The manuscript is consistent and logical enough even without these sentences. Rather than mentioning these, I suggest the authors discuss the optimal model, pros and cons, based on the situation.

(Response 4.5) Thanks for the suggestion. We agree. The sentences have been replaced as suggested.
è Ok. It is clearer. No more comment on this issue.

  1. In response to the reviewer’s additional comments

Point 5.1: Why did the authors separate training and test dataset in this way? Would it be better to combine all together, and then split it as training and testing data? Was there any specific purpose? For instance, was this model for Long-term experimental plots than NFRI permanent plots? If there is any additional information which can help readers understand, explain it in method section.

(Response 5.1) We appreciate your suggestions and comments on data splitting. Additional clarifications have been added in the materials and methods section (see Lines 176-185). As mentioned above, we agree that random selection of a subset from the combined dataset is a commonly-used data splitting approach. However, because the data were collected from two different sources, we think it is more appropriate to keep the data separated. The two fully independent datasets were also advantageous to evaluate the robustness of the methods/models. Although the number of the NTUEF long-term experimental plots is less than that of the NFRI permanent plots, more remeasurements per plot were taken. Thus, the NTUEF plots were more suitable in model/method fitting/training to capture the complete size-density relationship trajectories.
è Ok. It is clear. I suggest the authors note this issue in Discussion section. With the authors’ comment above, it is more logical and concrete if the authors write additional short comments about the caution of interpreting model fitting and validation, as I suggested this in Point 4.3.

Additional comments in the second rounding of peer-review

è Do not use both word for model and fitting (e.g. model/method, fitting/training). It is just redundant. Select one word for each expression. I am sure that all potential subscribers should understand the meaning. Since this manuscript includes both parametric and nonparametric methods, the author can use fitting & validation or train & test.

The authors already used “validation”, so I suggest “fitting” instead of “training”. I also recommend “model” without “method”.

Recommended alternatives:

model/method à model

fitting/trainingà fitting

Author Response

Journal: Forests

Manuscript ID: forests-789226

Title: Comparison of parametric and nonparametric methods for estimating size-density relationships in old-growth Japanese cedar plantations

Reviewer 2

In this second rounding of peer-review, my comments are blue color.

Authors’ responses are shown in green.

  1. In response to the reviewer’s overall comments

Point 1.1: As this article focuses on the size-density relationships in Japanese cedar planation of Taiwan, based on the scope of this journal, it is suggested for publication in this journal, Forests. As the author indicated in their manuscript, this kind of study has not been addressed commonly yet. Thus, the originality of this work lies in the comparison of parametric and nonparametric methods for size-density relationships especially in Japanese cedar plantation of Taiwan. This article would be of interest to forest modelers.

This manuscript was judged to be well-organized with clear research objectives and methodologies. Also, the data sources used in this study should be adequate to evaluate the model. The manuscript was generally readable with concise paragraphs and sentences. The author did compare the parametric and nonparametric methods. However, it may be improved by mentioning the practical point. There are several parts which can be improved by reconsidering and revising it. It is expected to be more concrete and logical after revision.

(Response 1.1) We greatly appreciate your effort and invaluable comments on this manuscript. We have carefully addressed all issues indicated in the review report and believe that the revision is suitable for further consideration for publication.

è Thank you for authors’ effort. I could feel the authors have carefully revised this manuscript. It was highly improved. Attached are my additional comments.

            è  We very much appreciate the reviewer’s comments!   

  1. In response to the reviewer’s suggestions in the Introduction

Point 2.1: The introduction provided adequate historical context and scientific rationale for this research. The research objectives were well supported by this logic. Line 59: However, if Japanese cedar is an important species, as referred by the authors, wasn’t there any study on size-density relationship for the species? If it exists, what would be the difference from the previous one? Or can you compare this study to Nishizono and Tanaka (2012)? I wonder about the features which can solve the current problem. It is needed to emphasize the necessity from the practical point of view. (even though the topic is interesting, will it be helpful for Japanese cedar to be managed?)

(Response 2.1) Thanks for the suggestion. More justification and clarification of this study were added in the Introduction. The practical implication of this study was also emphasized (see Lines 64-86).

            è Now it’s clear. No more comments on this issue.

III. In response to the reviewer’s suggestions in the Materials and Methods

Point 3.1 2.1.1. National Forest Resource Inventory (NFRI) permanent plots. Line 72: the plot size of 0.02 ha seems to be small. Assume that a stand has dbh 55.6cm, tph 340, age 93 according to the table 1, there might be less than 7 trees in plot of 0.02 ha. Will the number of sample tree be represented for a stand sample? Or there is no such a case in this data? It is better to explain about this.
(Response 3.1) Thanks for the comment. We agree that the plot size was not as large as some ecological long-term study plots, while the plot size is commonly used in forestry practice. The total number of sample trees ranged from 10 to 88 trees per plot at stand ages of 8-93 (yrs). (see Lines 105)

è I partially agree that 0.02 ha plot size is used in forestry practice. However, it would be rather suitable for young stands. It should not use for old and large stands. Fortunately, there were 10 trees per plot at least. Still, it is relatively small, but understandable.
The number of sample trees per plot is important for many statistical approaches. Thus, I wish the authors should be careful with this matter in future work.
The authors’ explanation was clear. no more comments on this issue.

è Thank you very much for bringing this up. We will be more careful to examine the number of sample trees per plot when dealing with the old-growth forests in the future. Appreciated it!

Point 3.2 Line 75: Wasn’t there any difference of silviculture practice among the offices? Or Did author just not consider this kind of effect even though the silviculture practice was quite different? It should be described more to explain the controlled variables.

(Response 3.2) Each office followed the identical standard of the silviculture practice regardless of sites or elevations. In addition to herbaceous vegetation removal and seedling replanting, no other treatments were applied on the permanent plots. The details have been added (see Lines 94-97)

            è The authors’ explanation was informative. It is a clear description. no more comments on this issue.

Point 3.3 Line 80: A criterion, plots with Japanese cedar basal area greater 30%, is quite low. How do we say this is a plantation? Are the authors sure that the censored sample plots are Japanese cedar plantation? If the other species are vigorous, it probably influences the size-density relationship a lot. Is there any other reference for this criterion, BA > 30%? In my opinion, it should be greater than 50% at least, especially because the data is originated from NFI.

(Response 3.3) Thanks for pointing out. More details on plot selection have been added (see Lines 101-105). Plots with a majority of poor-quality standing trees or dominated by other weedy species were not included in analyses. A total number of 515 observations were collected from 222 permanent plots where Japanese cedar basal area was greater than 30% of the total basal area and Japanese cedar dominated at least 20% of the main canopy. Among all observations, the total number of sample trees ranged from 10 to 88 trees per plot.

            è Ok. It is clearer. However, I suggest the authors discuss the model self-evaluation with this criterion. This factor can be dealt with one of the model limits.

è Thanks for pointing out. The limitation of the models has been added in Lines 317-320.

Point 3.4: 2.1.2. Long-term experimental plots. Line 98: Add the variable Dq, quadratic mean diameter, in table 1 which was used as a predictor.

(Response 3.4) Thanks for pointing out. Done. The variable Dq, quadratic mean diameter has been added.

è Ok. It is better. no more comments on this issue.

Point 3.5: 2.2.1. Methods section was also systematic and generally described the procedures well. However, the detail may not be enough even though the author cited accurate replication by experienced modelers, e.g. VanderSchaaf and Burkhart (2008). The method section should describe more about the model you selected. Line 120: What is the meaning of Reduced model? Why did you compare Reduced model with Full model? What information can readers get? What is this for? Application, biometric concept, or accurate estimation? Specify more about this. The explanation is not enough in the manuscript

(Response 3.5) More descriptions of the full model and reduced model were added in this section (see Lines 143-149).

è It is better. Thank you for authors’ effort. no more comments on this issue.

Point 3.6: 2.2.3. Random forest for regression. As described in the manuscript, Random forest takes variables at random for each of decision trees. However, there were only two independent variables considered in this study. Two independent variables are quite limited for random selection in random forest ensemble method. How about just decision tree and/or bagging? In addition, how did you tune the hyperparameters such as max. depth, min. samples leaf, and/or min. sample split as well as no. of decision trees? The results were overfitted probably because of this.

(Response 3.6) Thanks for the comments. The description of the selection of hyperparameters was added (see Lines 173-174). We agree that the random forest algorithm may not be powerful with the small number of predictors. However, since the random forest algorithm is the most popular method compared to other ensemble algorithms, the results of this study can add some additional insights of using this method. In the results and discussion section, more discussion has been added (see Lines 283-291).

è It is better. no more comments on this issue.

Point 3.7: 2.3. Method evaluation. Line 147-149: If the method of splitting data was not applied only for random forest, but also for Segmented regression models and Penalized spline, then it should be explained in the section of 2.3. Method evaluation

(Response 3.7) Clarification made (see Lines 176-188).

è It should be as it is. Now it is clear. no more comments on this issue.

  1. In response to the reviewer’s suggestion in the Results and Discussion

Point 4.1: Your Results section was well organized and objectively presented your data and analysis. However, it can be improved with the support of additional graph information. Line 168, Figure 2: The predicted line is short. Draw it a little bit longer so that all the data samples are estimated within the line, especially in Fig. 2a.

(Response 4.1) Thanks for catching this. The predicted lines were modified to cover the entire range of the data.

è Now it is clearer. no more comments on this issue.

Point 4.2: 3.1. Size-density relationships estimated by the segmented regression models. Definite comparison is not enough between two segmented regressions except for mentioning the parameter estimates. Which one is better? Which one can the authors recommend? Your discussion offered reasonable and logical explanations for your discoveries. In addition, many of points were upheld by pertinent citations.

(Response 4.2) Appreciate the comments! More discussion and suggestions were added (see Lines 235-243).

è Great. Now it is clearer. no more comments on this issue.

Point 4.3: 3.1. Size-density relationships estimated by the segmented regression models Line 217-223: I agreed to the author’s discussion. It may be because of the overfitting of random forest. Line 249-258: I agreed to the author’s discussion. To solve this problem, why didn’t you mix the two dataset and shuffle it for training and test data?

(Response 4.3) Thanks for the suggestion. We agree that it is common to randomly select a small portion of the data as a validation dataset. However, in this study, we think it is more appropriate to keep the data separated because the data were obtained from two different sources. The clarification has been given in Lines 176-185.

The two fully independent datasets can also be helpful to evaluate the robustness of the methods/models. We understand that from a statistical point of view, extrapolation from the original range can produce poor estimates but in forestry practice, the extrapolation could sometimes be required when implementing the models/methods due to the time and cost constraints. The discussion has been added as well (see Lines 285-291).

è Ok. I understand the data characteristics. I suggest the authors note this issue in Discussion section. With the authors’ comment above, it is more logical and concrete if the authors write additional short comments about the caution of interpreting model fitting and validation.

è Thanks for the suggestion. Additional information was added in the discussion (see Lines 288-290).

Point 4.4: line 266-268: It is better to insert a figure 3 with arithmetic N vs. arithmetic Dq, if the authors try to make readers understandable for actual self-thinning in dataset.

(Response 4.4) Thanks for pointing out. We rephrased the sentence to be more understandable.

è Ok. No more comment on this issue.

Point 4.5: line 273-276, 293-294: In my opinion, authors do not need to mention this kind of exceptional situation. I don’t think much this is the limitation of the present study. The manuscript is consistent and logical enough even without these sentences. Rather than mentioning these, I suggest the authors discuss the optimal model, pros and cons, based on the situation.

(Response 4.5) Thanks for the suggestion. We agree. The sentences have been replaced as suggested.
è Ok. It is clearer. No more comment on this issue.

  1. In response to the reviewer’s additional comments

Point 5.1: Why did the authors separate training and test dataset in this way? Would it be better to combine all together, and then split it as training and testing data? Was there any specific purpose? For instance, was this model for Long-term experimental plots than NFRI permanent plots? If there is any additional information which can help readers understand, explain it in method section.

(Response 5.1) We appreciate your suggestions and comments on data splitting. Additional clarifications have been added in the materials and methods section (see Lines 176-185). As mentioned above, we agree that random selection of a subset from the combined dataset is a commonly-used data splitting approach. However, because the data were collected from two different sources, we think it is more appropriate to keep the data separated. The two fully independent datasets were also advantageous to evaluate the robustness of the methods/models. Although the number of the NTUEF long-term experimental plots is less than that of the NFRI permanent plots, more remeasurements per plot were taken. Thus, the NTUEF plots were more suitable in model/method fitting/training to capture the complete size-density relationship trajectories.

è Ok. It is clear. I suggest the authors note this issue in Discussion section. With the authors’ comment above, it is more logical and concrete if the authors write additional short comments about the caution of interpreting model fitting and validation, as I suggested this in Point 4.3.

è Thanks for the suggestion. Additional information was added in the discussion (see Lines 288-290).

Additional comments in the second rounding of peer-review

è Do not use both word for model and fitting (e.g. model/method, fitting/training). It is just redundant. Select one word for each expression. I am sure that all potential subscribers should understand the meaning. Since this manuscript includes both parametric and nonparametric methods, the author can use fitting & validation or train & test.

The authors already used “validation”, so I suggest “fitting” instead of “training”. I also recommend “model” without “method”.

Recommended alternatives:

model/method à model

fitting/trainingà fitting

è Thanks for pointing out. The changes were made as suggested, which have been highlighted in the manuscript. The “model/method” has been changed to “model”, and the “fitting/training” has been changed to “fitting.”

Reviewer 3 Report

Authors have revised manuscript following the comments and current manuscript can be accepted for publication.

Author Response

Comments and Suggestions for Authors

Authors have revised manuscript following the comments and current manuscript can be accepted for publication.

Thanks to the reviewers' comments for making the manuscript more prefect.

Submission Date

15 April 2020

Date of this review

20 May 2020 12:25:09
